# Casein kinase II promotes piRNA production through direct phosphorylation of USTC component TOFU-4

Gangming Zhang [1], Chunwei Zheng [1], Yue-he Ding[1] & Craig Mello [1,2]

Piwi-interacting RNAs (piRNAs) are genomically encoded small RNAs that engage Piwi Argonaute proteins to direct mRNA surveillance and transposon silencing. Despite advances in understanding piRNA pathways and functions, how the production of piRNA is regulated remains elusive. Here, using a genetic screen, we identify casein kinase II (CK2) as a factor required for piRNA pathway function. We show that CK2 is required for the localization of PRG-1 and for the proper localization of several factors that comprise the 'upstream sequence transcription complex' (USTC), which is required for piRNA transcription. Loss of CK2 impairs piRNA levels suggesting that CK2 promotes USTC function. We identify the USTC component twenty-one-U fouled-up 4 (TOFU-4) as a direct substrate for CK2. Our findings suggest that phosphorylation of TOFU-4 by CK2 promotes the assembly of USTC and piRNA transcription. Notably, during the aging process, CK2 activity declines, resulting in the disassembly of USTC, decreased piRNA production, and defects in piRNA-mediated gene silencing, including transposons silencing. These findings highlight the significance of posttranslational modification in regulating piRNA biogenesis and its implications for the aging process. Overall, our study provides compelling evidence for the involvement of a posttranslational modification mechanism in the regulation of piRNA biogenesis.

Piwi-interacting RNAs (piRNAs) associate with Piwi-clade Argonautes in metazoans[1,2]. piRNAs regulate diverse biological processes including virus defense[3], sex determination[4], male fertility[5] and genome stability[2]. In *C. elegans*, piRNAs engage thousands of germline-expressed mRNAs[6] and are required for recognizing and silencing newly-introduced foreign transgenes[7].

In *C. elegans*, piRNAs are also known as 21 U-RNAs. 21 U-RNAs are transcribed by RNA polymerase II (Pol II) from thousands of genomic locations[8]. There are generally two types of 21 U-RNA loci, type 1 loci appear dedicated to the production of piRNAs and are located in huge clusters on chromosome IV (LGIV), while type 2 piRNAs are produced often bidirectionally at most pol II transcription start sites, including those producing longer transcripts such as mRNAs or ncRNAs. Both types of piRNAs are transcribed as capped short RNAs (piRNA precursors) of approximately 27 nts in length. Type 1 piRNA genes contain an 8 nt sequence element called the Ruby motif[9] located approximately 40 nts upstream of their transcription start sites. Expression of type 1 (but not type 2) piRNA precursors depends on components of the 'upstream sequence transcription complex (USTC)'—including PRDE-1, SNPC-4, TOFU-4 and TOFU-5[10]. After transcription piRNA precursors are trimmed by the PUCH complex which executes 5' end piRNA precursor cleavage[11]. The conserved exonuclease PARN-1 trims piRNA 3' ends from the piRNA precursors to generate the mature 21 nt piRNAs[12].

Casein kinase II (CK2) is a conserved serine/threonine kinase involved in many cellular processes including transcriptional

[1]RNA Therapeutics Institute, University of Massachusetts Chan Medical School, Worcester, MA 01605, USA. [2]Howard Hughes Medical Institute, Worcester, MA 01605, USA. e-mail: Craig.Mello@umassmed.edu

regulation, DNA repair and cell survival[13,14]. Phosphorylation mediated by CK2 affects protein-protein interaction, for example, the phosphorylation of Rap1 by CK2 promotes Rap1/Bqt4 interactions to facilitate telomere protein complex formation[15].

Aging in animals is associated with extensive functional decline, ranging from a decline in tissue integrity, motility, learning and memory, and immunity[16]. These changes are correlated with and likely caused at least partially by remodeling of the epigenome. However, the mechanisms underlying the loss of heterochromatin during the aging process remain largely unknown. Investigating the factors and mechanisms that contribute to the age-related loss of heterochromatin will provide valuable insights into the molecular basis of aging and its associated functional decline.

Here we show that CK2 is required for piRNA mediated silencing in *C. elegans*. Depletion of CK2 affects piRNA biogenesis. Furthermore, we demonstrate that CK2 is required for phosphorylation of the USTC component TOFU-4, which was previously reported to promote piRNA transcription[10]. We observed that this phosphorylation process is impaired during the aging process, resulting in disruptions to piRNA-mediated gene silencing. Collectively, our findings strongly indicate that CK2-mediated phosphorylation of TOFU-4 plays a critical role in regulating USTC assembly, promoting piRNA transcription, and is required for effective piRNA-mediated gene silencing during the aging process.

## Results

### CK2 promotes piRNA mediated silencing

In *C. elegans* extensive forward genetic screens have identified genes required for piRNA silencing. However, a limitation of these previous studies is that they require the resulting strains to be both piRNA silencing defective and viable. We reasoned that many important regulators of piRNA biogenesis might also function in pathways essential to development and viability. To identify such factors, we performed an RNAi-based screen of genes previously determined to be essential for viability. To do this we used a sensor system expressing a bright easily scored GFP transgene whose silencing requires an active piRNA pathway (Fig. 1a)[17]. In wild type worms, the sensor GFP signal is silenced within the pachytene zone of the ovary. However, in *Piwi* pathway mutants, bright GFP signal expands throughout the pachytene zone in the sensor animals (Fig. 1a, b).

Sensor animals were exposed to RNAi by feeding groups of about 100 worms on petri plates seeded with individual bacterial strains each expressing a dsRNA targeting one of 945 essential genes (Supplementary Data 1). This screen identified 39 positives, including several pathways that were previously described as essential for both viability and piRNA silencing (Supplementary Data 2). For example, the piRNA reporter was desilenced when *hda-1* and *smo-1* were depleted[17], as well as depletion of the integrator complex, which has been shown to terminate *C. elegans* piRNA transcription[18]. Furthermore, we found that other genes, such as splicing factors and nuclear pore complex components, were also capable of desilencing the piRNA reporter. Of these genes, we were particularly intrigued to discover that knockdown of components of the Cassien Kinase 2 (CK2) complex led to desilencing of the piRNA sensor. For example, we found that RNAi of *kin-10* (a regulatory subunit) and of *kin-3* (a catalytic subunit) desilenced the piRNA sensor (Fig. 1c). To confirm the RNAi findings, we used an auxin-inducible degron system to conditionally deplete KIN-3 and KIN-10. Using CRISPR we inserted sequences encoding 46 amino acids corresponding to the plant degron into the endogenous *kin-3* and *kin-10* genes in a strain expressing the plant F-box protein TIR1 which upon auxin binding associates with the DEGRON domain to recruit the proteosome machinery[19]. Exposure of this strain to auxin at 1 mM completely desilenced the piRNA sensor (Fig. 1b–d), and upon prolonged exposure caused lethality as expected.

We next examined whether piRNA mediated silencing of transposable elements is disrupted in CK2 mutants. We examined the *line2h* and *turmoil1* transposons, which are transposons silenced by piRNA pathway[20,21]. As expected, expression of the *line2h* and *turmoil1* transposon mRNAs were increased in *prg-1* and *kin-3* mutants (Fig. 1e). Together, these findings suggest that the CK2 complex promotes piRNA mediated silencing.

### CK2 promotes the levels of both mature and precursor piRNAs

We next wished to explore the effect of CK2 on piRNA levels. The levels of both mature and precursor piRNAs were determined as previously described[22] by deep sequencing small RNA populations from 2 biological replicates of *kin-3* degron animals after auxin exposure. For comparison we also analyzed small RNAs sequenced from wild type animals and from *prg-1* and *prde-1* mutants which are required respectively for stabilizing mature piRNAs and for promoting transcription of piRNA precursors. As previously shown[22], mature piRNAs were dramatically reduced compared to wild type levels in both *prg-1* and *prde-1* mutants (Fig. 2a, b) and were also dramatically reduced in *kin-3* depleted animals (Fig. 2a, b). Moreover, comparison of type 1 piRNAs in *kin-3* and *prg-1* mutants revealed similar levels of depletion across all -15,000 species (Supplementary Fig. 1). Examination of type 2 piRNA levels revealed that *kin-3* caused at most a partial reduction in levels, an effect similar to that of *prde-1*, which is required for transcription of type 1 but not type 2 piRNAs (Fig. 2c)[22].

Libraries prepared to enrich for the small capped RNAs that represent piRNA precursors revealed that piRNA precursor levels were reduced in the *kin-3* depleted libraries to levels similar to those found in *prde-1* mutants (Fig. 2d, e). Taken together, these findings suggest that CK2 is required for the production or stability of piRNA precursors.

### CK2 promotes the localization of USTC factors

Previous studies have shown that both PRG-1 protein levels and its localization to peri-nuclear nuage, P granules, are reduced in mutants that disrupt expression of piRNA precursors, while PRG-1 mRNA levels remain unaffected[23]. These findings suggest that PRG-1 like some other Argonaute proteins such as Ago2 becomes unstable when unloaded[24]. Consistent with these previous findings we found that in *kin-3* depleted animals, PRG-1 mRNA levels were not reduced compared to wild type (Supplementary Fig. 2a), while in contrast, GFP::PRG-1 failed to localize in P-granules and instead localized diffusely in the cytosol (Fig. 3a). We observed an identical change in GFP::PRG-1 localization in animals with mutations in *tofu-4* (Fig. 3a), a gene previously shown to be required for piRNA transcription and for PRG-1 protein stability[10]. When assayed by western blotting with GFP-specific antibodies in *kin-3* and *tofu-4* mutants, the band corresponding to full-length GFP::PRG-1 protein was strongly reduced and a 30KD band, corresponding in size to free GFP, became prominent (Fig. 3b). This finding suggests that the N-terminal GFP tag is proteolytically released from GFP::PRG-1 during the PRG-1 instability induced by *kin-3* and *tofu-4* mutations, and suggest that persistence of this proteolytic fragment explains the cytoplasmic GFP fluorescence observed in the mutants. We also monitored GFP::PGL-3, a P-granule marker whose localization in nuage does not depend on piRNA biogenesis (Supplementary Fig. 2b, c), and found that GFP::PGL-3 is localized properly in the nuage of *kin-3* depleted animals.

SNPC-4 and PRDE-1 are two USTC factors required for piRNA biogenesis in *C. elegans*[10]. In germline nuclei, they form foci that co-localize with the piRNA gene clusters on LGIV. We found that *kin-3*-depleted animals and as previously reported *tofu-4* mutants[10] failed to form subnuclear foci of SNPC-4::GFP and mCHERRY::PRDE-1 (Fig. 3c, d). These findings suggest that CK2 acts along with TOFU-4 to promote SNPC-4 and PRDE-1 localization to piRNA clusters where

transcription occurs. GFP::KIN-3 localizes broadly to somatic and germline nuclei where it appears enriched on chromatin, including within the pachytene region of the germline (Fig. 3e), a localization consistent with a possible role in piRNA expression. Indeed, we can see the colocalization between KIN-3 and PRDE-1. However, we did not detect any enriched signal of GFP::KIN-3 in PRDE-1 foci. Additionally, we did not identify any changes of GFP::KIN-3 localization in *tofu-4* mutants (Supplementary Fig. 2d, e).

## CK2 directly Phosphorylates USTC factor TOFU-4

The USTC is thought to bind the promoters of piRNA genes to drive their expression in the *C. elegans* germline. TOFU-4 is a component of the USTC and KIN-3 was identified as a yeast two-hybrid (Y2H) binding

partner of TOFU-4[10]. We therefore asked if CK2 interacts with TOFU-4. We found that KIN-3 co-immunoprecipitated TOFU-4 (Fig. 4a). Since CK2 is a conserved serine/threonine kinase, we examined whether CK2 can phosphorylate TOFU-4 in vitro. TOFU-4, KIN-3 and KIN-10 proteins were purified from *E.coli* (Supplementary Fig. 3a), and in vitro kinase assays were performed. Incubation of KIN-3/KIN-10 (CK2 complex) with TOFU-4 resulted in a strong phosphorylation signal in in vitro phosphorylation assays (Fig. 4b), but KIN-3/KIN-10 failed to phosphorylate the Glutathione S-transferase (GST) protein (Fig. 4d). Thus, CK2 directly and specifically phosphorylates TOFU-4. Phosphorylation was dependent on CK2 activity, as inhibition of CK2 activity by addition of 4,5,6,7-tetrabromobenzotriazole (TBB), a CK2 inhibitor, reduced phosphorylation (Fig. 4b). A *C. elegans* phosphoproteome dataset[25] identified

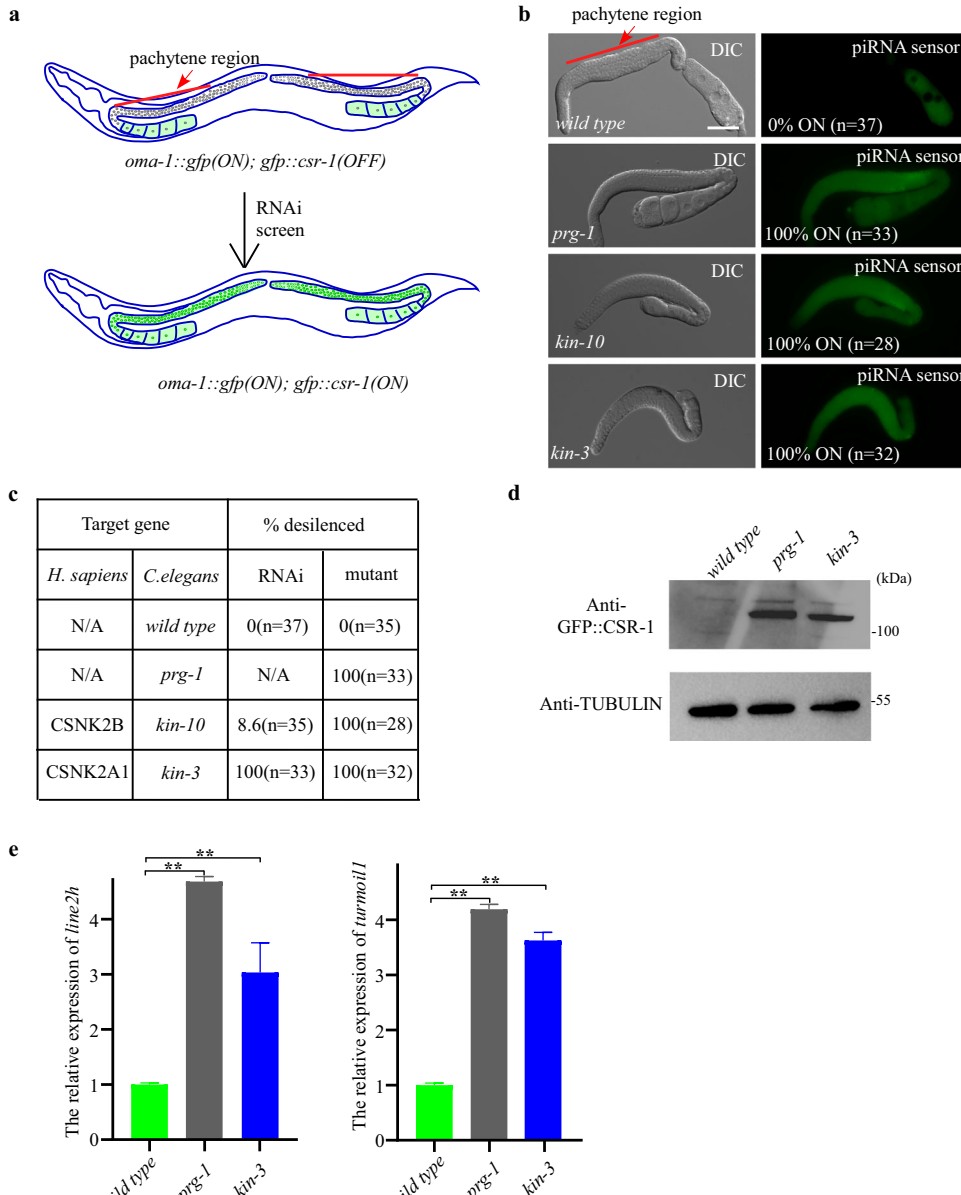

**Fig. 1 | CK2 complex promotes piRNA mediated silencing. a** Schematic overview of the piRNA sensor screen, the piRNA sensor contains an actively expressed OMA-1::GFP transgene and a GFP::CSR-1 transgene silenced by the piRNA pathway[37]. Inactivation of piRNA pathway will result in the GFP::CSR-1 expression in the perinuclear region. **b** The GFP::CSR-1 transgene is silenced in wild type worms, while in *prg-1*, *kin-10* and *kin-3* mutants, GFP::CSR-1 is expressed. **c** Percentage of worms with expressed piRNA sensors in wild type, *prg-1*, *kin-10* and *kin-3* mutants. n. Total number of animals scored. **d** Western results of GFP::CSR-1 protein levels in wild type, *prg-1* and *kin-3* mutants. One representative out of 3 independent experiments is shown. **e** qRT-PCR results of *line2h* and *turmoil* RNA levels in wild type, *prg-1* and *kin-3* mutants. mRNA level in wild type worms is set to 1.0. Data are shown as mean ± SD. *n* = 4 independent biological replicates; *p* values from two-tailed Student's t test. **\*\****p* < 0.01. Scale bars: 50 µm for **b**. Source data are provided as a Source Data file.

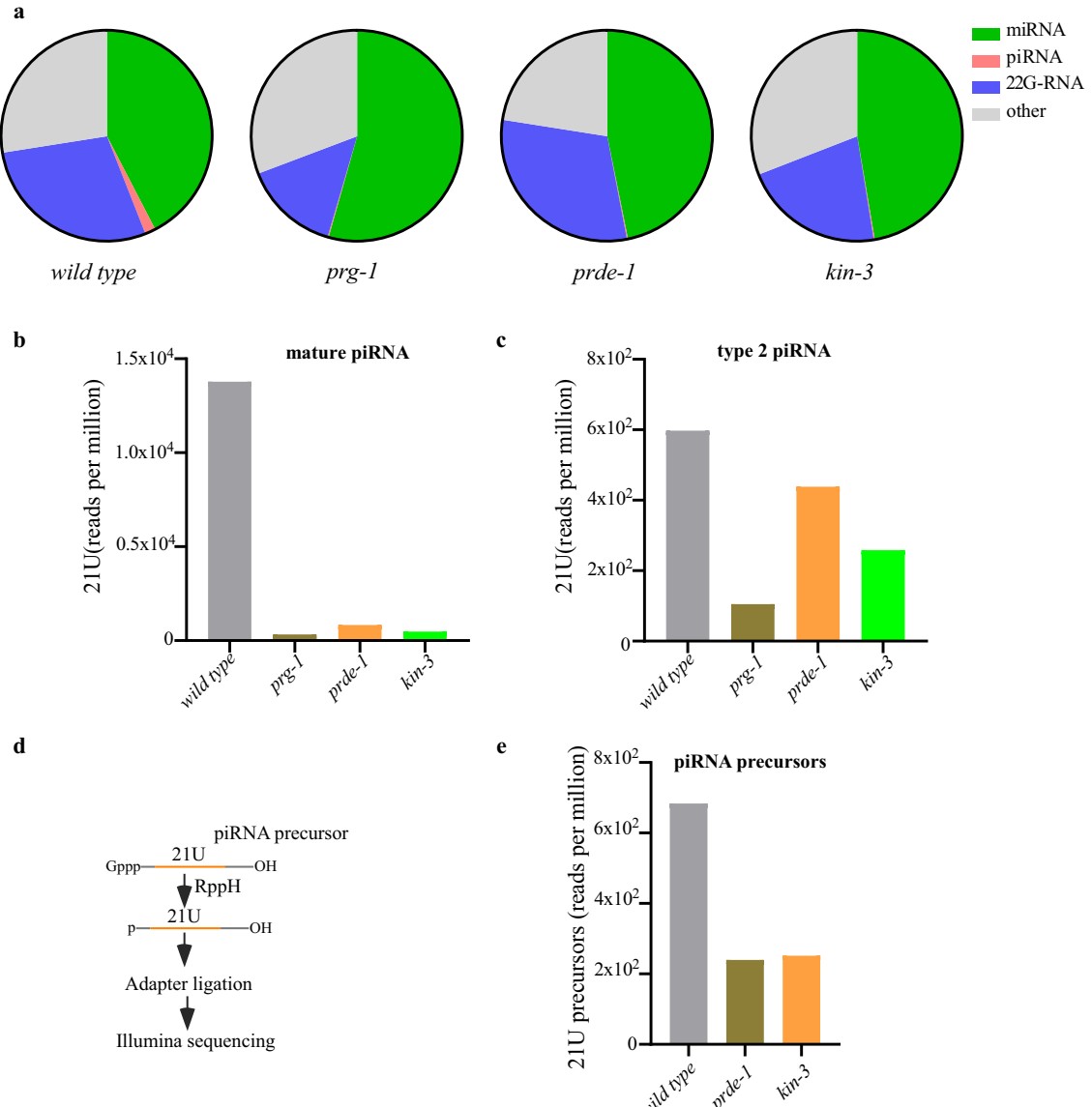

**Fig. 2 | CK2 promotes the levels of both mature and precursor piRNAs. a** The distribution of reads that correspond to genome annotation in small RNA libraries from one representative example of wild type, *prg-1*, *prde-1*, and *kin-3* mutants was depicted using pie charts. **b** Bar diagram displaying piRNAs mapping perfectly to known *C. elegans* piRNA loci in wild type, *prg-1*, *prde-1* and *kin-3* mutants. **c** Bar diagram displaying type 2 21U-RNA abundance in wild type, *prg-1*, *prde-1*, and *kin-3* mutants. **d** Outline of strategy to detect the piRNA precursors by deep sequencing. **e** Bar diagram displaying the number of piRNA precursors mapping to piRNA sequences in wild type, *prde-1*, and *kin-3* mutants. piRNA precursors from loci as above were examined. Small RNAs are pretreated with 5′ Pyrophosphohydrolase (RppH, NEB). Source data are provided as a Source Data file.

three phosphorylation sites in TOFU-4, each containing a consensus CK2 phosphorylation motif ([S/T]-X-X-[D/E]) (Fig. 4c). We found that synthetic TOFU-4 peptides containing either serine 131 (S131) or serine 229 (S229), were phosphorylated by CK2 in vitro (Fig. 4d). While a peptide containing serine 92(S92) was not phosphorylated. Using AlphaFold, we predicted the structure of TOFU-4 and discovered that S131 and S229 are in a disordered region and relatively conserved in nematodes (Supplementary Fig. 3b, c). Consistent with the idea that S131 and S229 are the predominant phosphorylation sites, a mutation of both the S131 and S229 to alanine in a bacterially-expressed full-length TOFU-4 protein greatly reduced phosphorylation of TOFU-4 in an in vitro CK2 kinase assay (Fig. 4b). In some instances, phosphorylation of a protein can cause a shift in mobility on an SDS gel[26]. To ask if phosphorylation of TOFU-4 by CK2 in vitro, or in vivo alters TOFU-4 mobility we used the previously published phos-tag gel method[27].

However, no significant shift was observed even for the in vitro phosphorylated TOFU-4 (Supplementary Fig. 4a).

To determine whether CK2 phosphorylation of TOFU-4 promotes piRNA silencing, we used CRISPR to introduce phosphoacceptor mutants in TOFU-4. As expected, based on the lack of phosphorylation in in vitro kinase assays, S92A mutants had no defect in piRNA silencing (Fig. 4e). The S131A and S229A lesions, on the other hand, each caused partial de-silencing individually, and complete desilencing when combined in *tofu-4(S131A S229A)* double mutant worms (Fig. 4e). We also generated *tofu-4(S131D)*, *tofu-4(S229D)*, and *tofu-4(S131D S229D)*, in which the potential phosphoacceptor residues were replaced with aspartic acid, a negatively charged residue that can sometimes mimic the phosphorylated state of a protein. We found that aspartic acid single and double mutants failed to preserve TOFU-4 function, causing de-silencing of

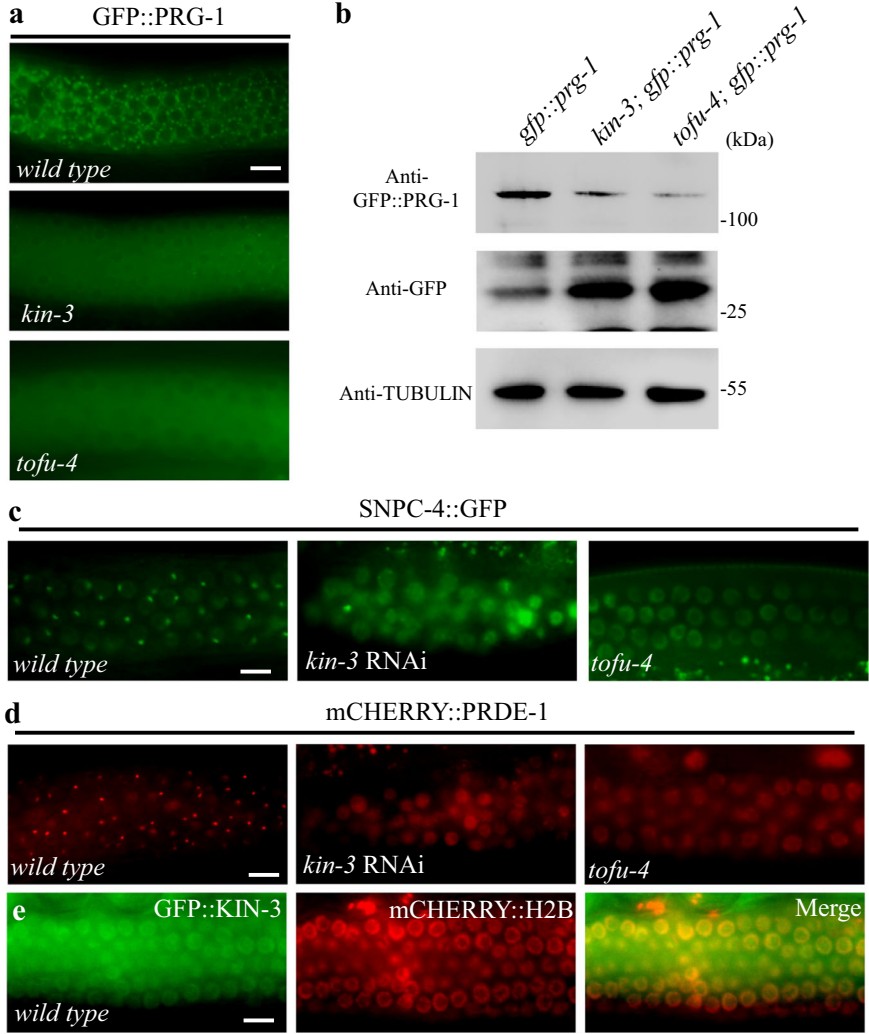

**Fig. 3 | CK2 promotes the localization of USTC factors. a** In wild type worms, GFP::PRG-1 is expressed in the perinuclear region, while in *kin-3* and *tofu-4* mutants, there is some diffused GFP signal. **b** Immunoblotting assays revealed that protein levels of GFP::PRG-1 are decreased while protein levels of free GFP are increased in extracts of *kin-3* and *tofu-4* mutants compared to those of wild type worms. One representative out of 3 independent experiments is shown. Compared to wild type worms, the SNPC-4::GFP (**c**) and mCHERRY::PRDE-1 (**d**) foci in *kin-3* and *tofu-4* mutants disappear. **e** GFP::KIN-3 in the nucleus is colocalized with the chromatin marker mCherry:H2B in young adult germ nuclei. Scale bars: 10 μm in **a**, **c**, **d** and **e**, for **a**, **c**, **d** and **e**, more than 50 worms are counted, the phenotype is similar. Source data are provided as a Source Data file.

the piRNA reporter, although at intermediate levels (Supplementary Fig. 4b), suggesting that the aspartic acid substitutions fail to precisely mimic the activities required to fully restore TOFU-4 function in piRNA silencing.

To ask if CK2 mediated phosphorylation of TOFU-4 affects piRNA production, we performed small RNA sequencing. We observed that the levels of both mature piRNAs and of piRNA precursors were reduced in the *tofu-4(S131A S229A)* mutants (Fig. 4f, g). Collectively, these findings suggest that CK2-mediated phosphorylation of TOFU-4 promotes piRNA production.

### CK2 Phosphorylation of TOFU-4 promotes USTC assembly

To explore the stability and protein interactions of the TOFU-4 phospho-acceptor mutants we performed Western blot and Co-IP studies. PRDE-1 and TOFU-4 protein levels appeared wild type, as measured by Western blotting, in *tofu-4(S131A S229A)* mutants (Fig. 5a). To further characterize the effect of these mutations on USTC assembly, we examined their effects on TOFU-4/PRDE-1 protein-protein interactions in vivo. We found that PRDE-1 co-precipitated with wild type TOFU-4 (Fig. 5b). However, TOFU-4(S131A S229A) did not co-precipitate PRDE-1

while TOFU-4(S131D S229D) exhibited decreased levels of co-precipitation (Fig. 5b).

In wild type worms, PRDE-1 and TOFU-4 form subnuclear foci in germline cells and colocalize with each other (Fig. 5c). We therefore examined whether and how the putative phospho-acceptor lesions affected their localization. Introduction of TOFU-4(S131A S229A) in an otherwise wild type strain caused mCHERRY::PRDE-1 and GFP::TOFU-4 to no longer co-localize and to instead become diffusely localized in nuclei (Fig. 5d), while in *tofu-4(S131D S229D)* mutants, mCHERRY::PRDE-1 and GFP::TOFU-4 are still co-localized in nuclear foci, albeit with smaller foci (Fig. 5e). Taken together these data suggest that phosphorylation of TOFU-4 by CK2 promotes the co-assembly of TOFU-4 with PRDE-1 and their co-localization at sites of piRNA transcription in *C. elegans*.

### Reduced CK2 activity contributes to piRNA mediated gene silencing defects in the aging process

Lastly, we monitored CK2 localization and activity in the aging process since a previous study showed that downregulation of CK2 activity promotes the expression of age-related biomarkers in *C. elegans*[28].

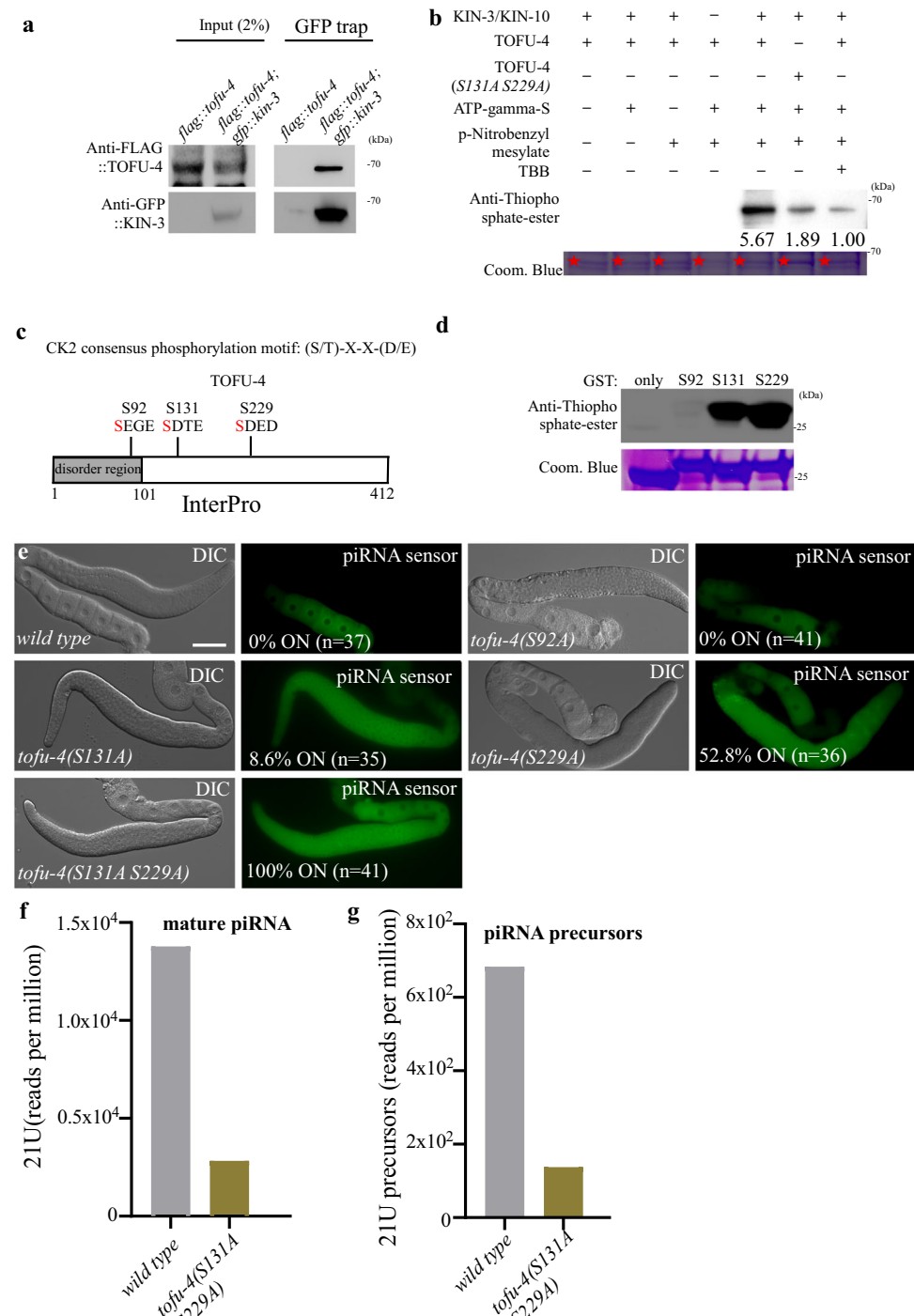

**Fig. 4 | CK2 Phosphorylates USTC factor TOFU-4. a** In co-IP assays, endogenous GFP::KIN-3 specifically immunoprecipitated endogenous FLAG::TOFU-4. One representative out of 2 independent experiments is shown. **b** Phosphorylation of TOFU-4 by the CK2 complex in an in vitro phosphorylation assay. Phosphorylation was detected by an antibody that specifically detects the thiophosphate-ester. The purified TOFU-4 (indicated by asterisks) was used in the assay. Coom. Blue, Coomassie staining. Quantifications of the phosphorylation are also shown. The phosphorylation level in the last lane is set to 1.0. One representative out of 3 independent experiments is shown. **c** Three phosphorylation sites were identified in TOFU-4, all of which are located within CK2 recognition motifs from a *C. elegans* phosphoproteome dataset. **d** In vitro phosphorylation assay showed that Serine 131 and Serine 229 were phosphorylated by CK2. TOFU-4 peptides composed of 20 residues flanking each putative CK2 phosphorylation site were used in the assay. Coom. Blue, Coomassie staining. One representative out of 3 independent experiments is shown. **e** The GFP::CSR-1 transgene is silenced in wild type and in *tofu-4(S92A)* worms, while in *tofu-4(S131A), tofu-4(S229A)* and *tofu-4(S131A S229A)* mutants worms, GFP::CSR-1 is expressed. **f** Bar diagram displaying piRNAs mapping perfectly to known *C. elegans* piRNA loci in wild type and *tofu-4(S131A S229A)* mutants. **g** Bar diagram displaying the number of piRNA precursors mapping to piRNA sequences in wild type and *tofu-4(S131A S229A)* mutants, small RNAs are pretreated with 5′ Pyrophosphohydrolase (RppH, NEB). Scale bars: 50 μm for **e** Source data are provided as a Source Data file.

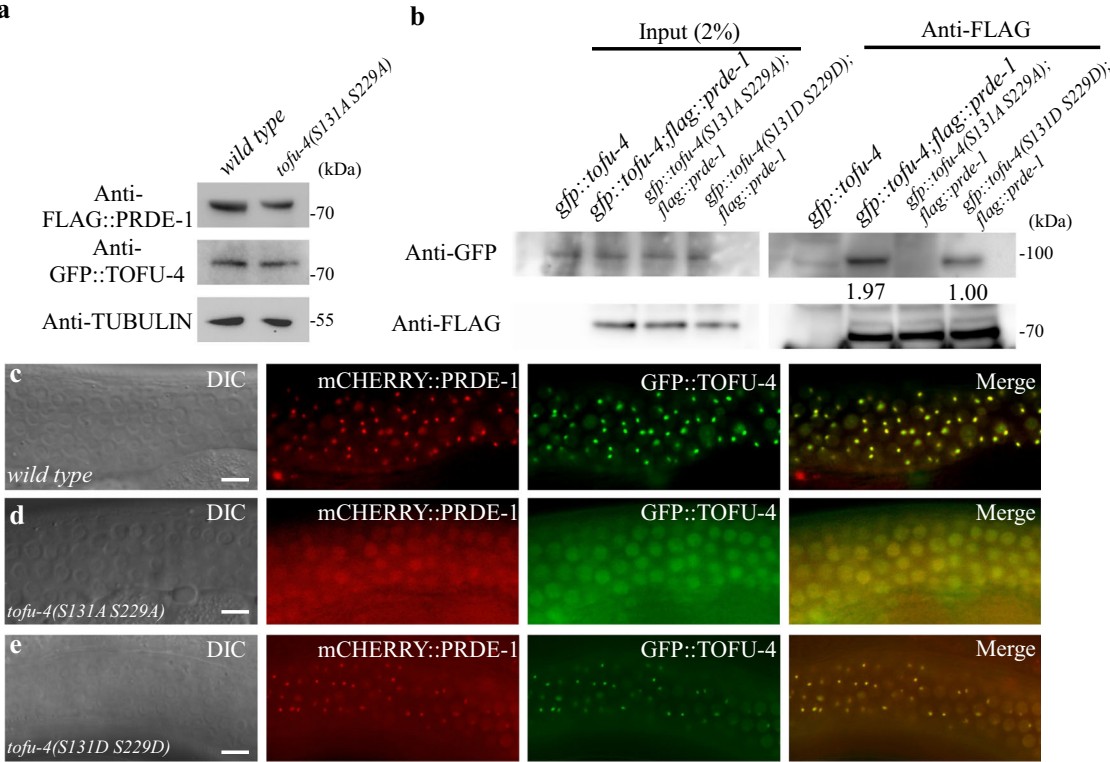

**Fig. 5 | CK2 mediated Phosphorylation of TOFU-4 promotes USTC assembly.**
**a** Protein levels of GFP::TOFU-4 and FLAG::PRDE-1 in wild type and in *tofu-4(S131A S229A)* mutants. One representative out of 3 independent experiments is shown.
**b** Western blot analysis GFP::TOFU-4 in FLAG::PRDE-1 immunoprecipitants from wild type, *tofu-4(S131A S229A)* and *tofu-4(S131D S229D)* mutants. One representative out of 2 independent experiments is shown. Compared to the wild type worms (**c**),

USTC factors GFP::TOFU-4 and mCHERRY::PRDE-1 fail to form nuclear foci in the germline in *tofu-4(S131A S229A)* mutants worms (**d**), and form smaller nuclear foci in the germline in *tofu-4(S131D S229D)* mutants worms (**e**). Scale bars: 20 μm (**c**, **d** and **e**), for **c**, **d** and **e**, more than 50 worms are counted, the phenotype is similar. Source data are provided as a Source Data file.

We examined the localization of GFP::KIN-3, which showed widespread expression, with a stronger presence in the nucleus. However, we found that during the aging process, the distinct pattern diminished (Fig. 6a). Subsequently, we investigated if the catalytic activity of CK2 in *C. elegans* was influenced in the aging process. The phosphotransferase activity of CK2 exhibited a 70% decline in worms at day 7 as compared to those at day 1 (Fig. 6b and Supplementary Fig. 4c). To further investigate the consequences of reduced CK2 activity, we examined the localization of USTC components. Consistent with our hypothesis, we observed the disappearance of GFP::TOFU-4 and mCHERRY::PRDE-1 foci (Fig. 6c, d). We also investigated the localization of GFP::PRG-1 and noted the loss of its perinuclear pattern (Fig. 6e). Moreover, western blotting analysis revealed a substantial reduction in the band corresponding to full-length GFP::PRG-1 protein in worms at day 7 (Fig. 6f). Consistently, we found the mature piRNA and the piRNA precursor levels were reduced in the aging process (Fig. 6g, h). Finally, we assessed whether the aging process affected our piRNA reporter and transposable elements. Our findings indicated significant desilencing of the piRNA reporter and notable increases in the *line2h* and *turmoil1* transposons (Fig. 6i–k). Taken together, these results are consistent with the possibility that declining CK2 activity contributes to the piRNA-mediated gene silencing defects observed during the aging process.

## Discussion

### A sensitized reporter to screen mutants involved in piRNA mediated gene silencing

In diverse metazoans piRNAs play maintain genome integrity by silencing transposons and have also been implicated in the transgenerational regulation of gene expression[29]. While significant progress has been made in understanding piRNA biogenesis and function, many

gaps in understanding persist, especially around elements of the pathway that are shared by other essential germline pathways. Here we constructed a sensitized reporter strain and used it to screen a library of 945 known essential *C. elegans* genes for RNAi-induced loss of piRNA silencing phenotypes. This screen identified 39 positive hits including previously identified genes required for piRNA biogenesis and function, including components of chromatin remodeling complexes, splicing factors, nuclear export factors, and transcription factors. Our screen identified about 20 genes not previously known to function in the piRNA pathway and here we have focused on characterizing one of these, *kin-10*, which along with *kin-3* comprise components of the CK2 kinase complex.

### TOFU-4 phosphorylation by CK2 promotes piRNA mediated gene silencing

The biogenesis of piRNAs involves a complex series of events, including transcription, processing, and loading into piRNA effector complexes. Previous studies have shown that USTC containing PRDE-1, SNPC-4, TOFU-4, and TOFU-5 is required for the expression of piRNA precursors[10]. Yet, little is known about how piRNA biogenesis is regulated. Here we have shown that the *C. elegans* casein kinase II, whose major subunits are encoded by the *kin-3* and *kin-10* genes, is required for piRNA mediated silencing. Mechanistically, our findings suggest that CK2 functions upstream of USTC assembly. TOFU-4 is a direct CK2 substrate and mutation of presumptive CK2 phosphorylation sites in TOFU-4 did not affect its interaction with KIN-3 (Supplementary Fig. 4d), but instead reduced the association of TOFU-4 with the USTC component PRDE-1. In the CK2 mutants USTC failed to assemble and piRNA biogenesis was defective (Fig. 7). A search within the phosphorylation database[25] focusing on the USTC factors (PRDE-1, SNPC-4,

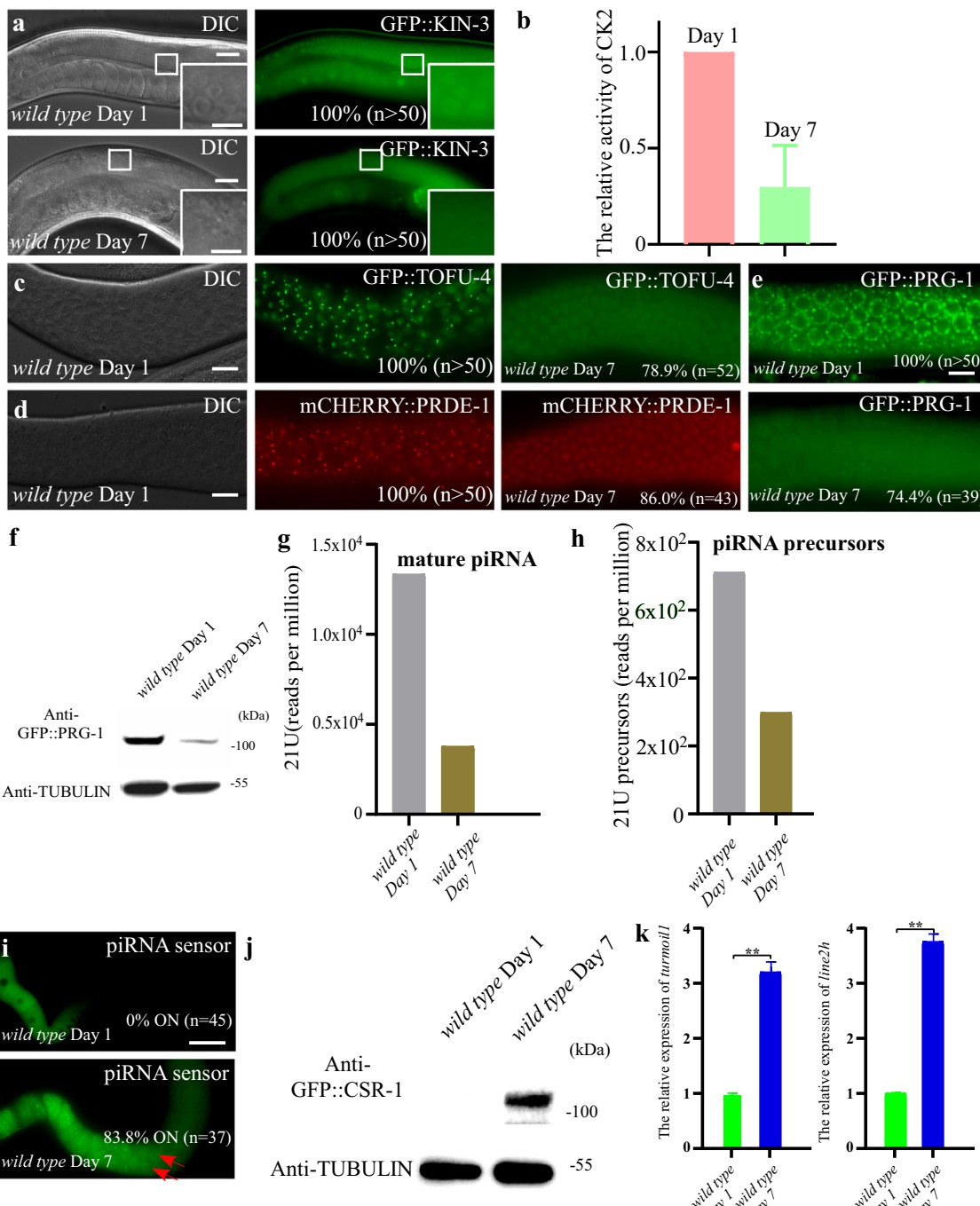

**Fig. 6 | Reduced CK2 activity in the aging process underlies piRNA mediated gene silencing defects. a** Subcellular localization of GFP::KIN-3 in wide type Day 1 and wild type Day 7 worms. n. Total number of animals scored. **b** The activity of CK2 to phosphorylate purified TOFU-4 peptide is decreased in the aging process. Lysates from wide type Day 1 and wild type Day 7 worms of adulthood were used in in vitro phosphorylation assays. The relative activity of CK2 from wide type Day 1worms is set to 1.0. Data is shown as mean ± SD. *n* = 3 independent biological replicates. Compared to wild type Day 1 worms, the nuclear foci of USTC factors GFP::TOFU-4 (**c**) and mCHERRY::PRDE-1 (**d**) in wild type Day 7 worms disappear. n. Total number of animals scored. **e** GFP::PRG-1 becomes diffused in wild type Day 7 worms compared to those of wild type Day 1 worms. n. Total number of animals scored. **f** Immunoblotting assays revealed that protein levels of GFP::PRG-1 are decreased in extracts of wild type Day 7 worms compared to those of wild type Day 1 worms. One representative out of 3 independent experiments is shown. **g** Bar

diagram displaying piRNAs mapping perfectly to known *C. elegans* piRNA loci in wild type Day 1 and wild type Day 7 worms. **h** Bar diagram displaying the number of piRNA precursors mapping to piRNA sequences in wild type Day 1 and wild type Day 7 worms. Small RNAs are pretreated with 5' Pyrophosphohydrolase (RppH, NEB). **i** Percentage of worms with expressed piRNA sensors in wide type Day 1 and wild type Day 7 worms. n. Total number of animals scored. **j** Western results of GFP::CSR-1 protein levels in wide type Day 1 and wild type Day 7 worms. One representative out of 3 independent experiments is shown. **k** qRT–PCR of *turmoil1* and *line2h* transposon levels in wide type Day 1 and wild type Day 7 worms relative to *act-3* mRNA are displayed. mRNA level in wild type worms is set to 1.0. Data is shown as mean ± SD. *n* = 4 independent biological replicates; *p* values from two-tailed Student's t test. **p* < 0.01. Scale bars: 20 μm (**a** and **c**–**e**), 50 μm for **i** and 10 μm (insets in **a**). Source data are provided as a Source Data file.

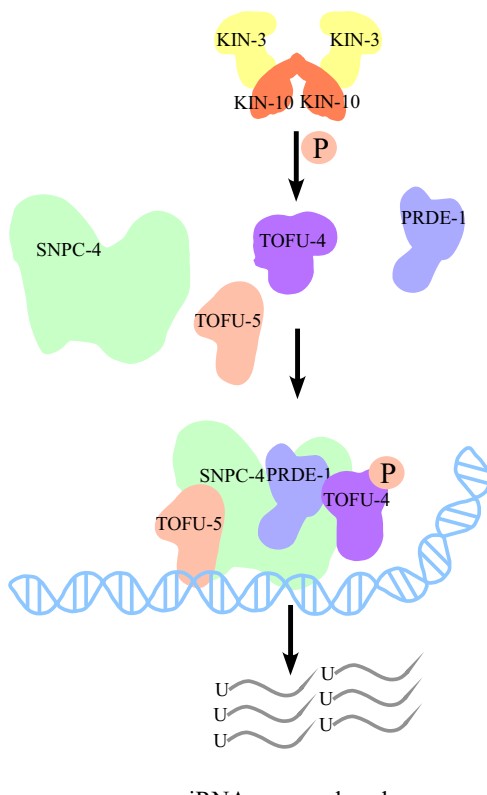

**Fig. 7 | CK2 mediated phosphorylation of TOFU-4 promotes piRNA mediated silencing.** A proposed model representing the regulation of phosphorylation of TOFU-4 by CK2 promotes USTC assembly and piRNA biogenesis in the germline. The production of piRNAs silence the foreign transgenes and transposons.

TOFU-4 and TOFU-5), in addition to the recently identified chromodomain factor UAD-2 which localizes to piRNA cluster chromatin[30], revealed that among these factors, TOFU-4 is the sole protein displaying detectable phosphorylation. While there could be additional piRNA biogenesis factors like UAD-2 subject to CK2 phosphorylation, our study suggests that posttranslational modification of TOFU-4 by CK2 is important for assembly of the piRNA biogenesis machinery and for piRNA pathway function.

Our findings suggest that phosphorylation of TOFU-4 promotes its association with PRDE-1 and the co-assembly of the two proteins on piRNA cluster genes where along with other components of the USTC they promote piRNA transcription. In *S. pombe*, CK2 directed phosphorylation of a component of the shelterin telomere protein complex, Rap1, promotes interactions with factors that regulate proper telomere tethering to the inner nuclear envelop and the formation of the silenced chromatin structure at chromosome ends[15]. In the *C. elegans* pachytene germline PRDE-1 foci colocalize with large regions of chromosome IV where the majority of piRNA genes reside and become positioned near the nuclear periphery[22]. The transcription of piRNA genes is thought to occur within chromatin marked by H3K27me3, a mark also found in telomeric chromatin[31]. It will be interesting to learn whether CK2 phosphorylation of TOFU-4 promotes the co-localization of the huge megabase-scale clusters of piRNA genes perhaps to coordinate piRNA transcription with piRNA export and processing.

### Aging and piRNA mediated gene silencing
During the aging process, loss of heterochromatin is a consistent trend in species ranging from humans to fungi[32]. How and why

heterochromatin is lost in the aging process is not known and the mechanisms are complex and generally cell-type specific. Since piRNA pathways promote the maintenance and installation of heterochromatin in diverse animal germlines it is possible that CK2 promotes heterochromatin maintenance, at least in part, through the maintenance of piRNA activity. Considering that heterochromatin maintenance is implicated in lifespan regulation[32], further work is needed to explore all the possible relationships between CK2 activity, piRNAs and lifespan.

In summary, our work has uncovered a role for CK2 in the regulation of piRNA biogenesis in *C. elegans* and provides a starting point for a more detailed examination of the role of post-translational modifications in piRNA biogenesis in the aging process.

## Methods
### *C. elegans* Strains and Bacterial Strains
All the strains in this study were derived from Bristol N2 and cultured at 20 °C on Nematode Growth Media (NGM) unless otherwise indicated[33]. The strains used in this study are listed in Supplementary Data 3.

### RNAi screen
RNAi screen was performed against all 945 genes in the embryo lethal subset of the *C. elegans* RNAi collection (Ahringer). Synchronized L1 animals of the reporter strain were plated onto RNAi feeding plates with about 100 worms per plate and were grown at 20 °C. The desilencing phenotype was scored when the worms grew to L4 or young adult stage.

### CRISPR/Cas9 genome editing
For CRISPR/Cas9 experiments, Cas9 ribonucleoprotein (RNP) editing was used to generate the CRISPER lines[34]. Cas9 genome editing mixture containing Cas9 protein (0.5 µl of 10 µg/µl), two crRNAs (each 1.4 µl of 0.4 µg/µl), annealed PCR donor (25 ng/µl), and PRF4::*rol-6(su1006)* plasmid (0.8 µl of 500 ng/µl) was incubated at 37 °C for 15 mins before injecting animals. Guide RNA sequences and donors used in this study are listed in Supplementary Data 4.

### RNAi inactivation experiments in *C. elegans*
For RNAi inactivation, single-stranded RNAs (ssRNAs) were transcribed from SP6-flanked PCR templates. ssRNAs were then annealed and injected into animals carrying various reporters. Injected animals were placed on fresh plates overnight. The progeny was analyzed.

### In vitro pull-down assays
cDNAs encoding full-length or mutated TOFU-4 was cloned into pET-28a (for His tagging) or pGEX-6P-1 (for GST fusion). GST fusion proteins were incubated with His-tagged proteins and Glutathione beads (for GST fusion proteins). Bound proteins were analyzed by Western blot using an anti-His antibody. 30% of the fusion protein used for pull-down is shown as input.

### Auxin treatment
The auxin-inducible degron system was as described[35]. The degron-tagged worms were placed on NGM plates with 1 mM indole-3-acetic acid (IAA; Alfa Aesar, 10171307), and then collected for further analysis.

### Co-IP and western blotting
Synchronous adult worms were collected and washed three times with M9 buffer. The worms were then homogenized in a FastPrep-24 benchtop homogenizer (MP Biomedicals). Worm extracts were centrifuged at 14000 × *g* for 15 mins at 4 °C. The worm extracts were incubated with anti-FLAG or GFP trap beads for 2 hrs at 4 °C on a rotating shaker. The beads were washed three times and subjected to SDS-PAGE. Signals were detected using corresponding primary and secondary antibodies.

## Small RNA cloning and data analysis

The small RNA cloning was conducted as the previous paper[22]. Total RNAs were extracted by Trizol (Sigma Alrich) and small RNAs were enriched by a mir-Vana miRNA isolation kit (Thermo Scientific). For mature piRNA sequencing, samples were treated with homemade PIR-1 for 2hs. For piRNA precursor sequencing, samples were pretreated with 5' Pyrophosphohydrolase (RppH, NEB) for 1 h, and purified with RNA purification kit, then treated with homemade PIR-1 for 2hs. The small RNAs were then ligated to a 3' adaptor (5' rAppAGATCGGAA-GAGCACACGTCTGAACTCCAGTCA/3ddC/3'; IDT) by T4 RNA ligase 2(NEB). The 5' adaptor containing 6 nt barcode was ligated using T4 RNA ligase 1. The ligated products were reverse transcribed using SuperScript III (Thermo Fisher Scientific). The cDNAs were amplified by PCR and the libraries were sequenced using the HiSeq systems (Illumina) platform at the UMass Medical School Deep Sequencing Core Facility.

The small RNA sequencing data were analyzed as described[22].

Briefly, adaptors were trimmed from fastq files. Reads were trimmed to retain only 18–30 nts. The reads were mapped to piRNAs or piRNAs precursors with an arbitrary length of overhanging sequence either side of the 21U sequence using the exact matches. Reads were normalized to the total reads mapped to the genome.

## Protein expression and purification

All genes were amplified by PCR and cloned into the pET-28a vector to produce His6-tag-fused recombinant proteins, or the pGEX-6P-1 vector to produce GST-tag-fused recombinant proteins. All recombinant proteins used in this study were expressed in *E. coli* BL21-CodonPlus (DE3) induced by 0.3 mM IPTG for 16 h at 20 °C and collected by sedimentation.

To purify His-KIN-3, His-KIN-10, His-TOFU-4, HIS-TOFU-4(S131A S229A) GST-KIN-3 and GST-tagged TOFU-4 fragments, the *E. coli* cells were resuspended in binding buffer (50 mM Tris-Cl pH 7.9, 500 mM NaCl and 10 mM imidazole), lysed with a high-pressure homogenizer and sedimented at 18000 rpm for 30 min at 4 °C. The supernatant lysates were purified on Ni-NTA agarose beads (for His6-tagged proteins, Qiagen) or Glutathione High Capacity Magnetic Agarose Beads (for GST-tagged proteins, SIGMA). After 2 extensive washing with binding buffer, the proteins were eluted with His6 elution buffer (50 mM Tris-Cl pH 7.9, 500 mM NaCl and 500 mM imidazole, for His6-tagged proteins) or GST elution buffer (50 mM Tris-Cl pH 7.9, 500 mM NaCl and 10 mM GSH, for GST-tagged proteins). All the eluted proteins were concentrated by centrifugal filtrations (Millipore), loaded onto desalting columns (GE Healthcare), then eluted with PBS buffer (140 mM NaCl, 2.7 mM KCl, 10 mM $Na_2HPO_4$, and 1.8 mM $KH_2PO_4$). The eluted proteins were stored in aliquots at -80 °C.

## In vitro phosphorylation assays using ATP-γS

The in vitro phosphorylation assays using ATP-γS was conducted as the previous paper[36]. To detect whether a protein is directly phosphorylated by CK2 complex, the His-KIN-3 and His-KIN-10 proteins purified from *E. coli* were used for the kinase assay. Recombinant His-TOFU-4, GST, GST-TOFU-4(S92), GST-TOFU-4(S131) and GST-TOFU-4(S229) proteins purified from *E. coli* were used as substrates. The reaction mixtures containing 500 μM ATP-γS were incubated for 1 h at 30 °C in kinase assay buffer (30 mM HEPES, 50 mM potassium acetate, 5 mM $MgCl_2$). For detection by anti-thiophosphate-ester antibody, the reaction mixtures were further supplemented with 2.5 mM *p*-nitrobenzyl mesylate (PNBM). The alkylating reactions were allowed to proceed for 1 h at 25 °C. The reaction systems were terminated with SDS sample buffer and boiled before analysis by SDS-PAGE.

To detect CK2 activity in worms, the assay was conducted following a previously described protocol[28]. Briefly, worms were lysed, the protein concentration in supernatant was measured and adjusted. The in vitro phosphorylation assay was conducted using ATP-γS. GST-TOFU-4(S229) proteins purified from *E. coli* were used as substrates The reactions were initiated by adding worm lysates and incubated for 30 min.

## Reagents and kits

Refer to Supplementary Data 5 for details.

## Statistical analysis

GraphPad Prism 7 or Microsoft Excel was used for all statistical analysis. All data are shown as mean ± SD. Unpaired two tailed t tests were performed for statistical analysis. a *P*-value less than 0.01 was considered significant (\*\*).

## Reporting summary

Further information on research design is available in the Nature Portfolio Reporting Summary linked to this article.

## Data availability

The data supporting the findings of this study are available from the corresponding authors upon request. The small RNA sequencing data is available from the NCBI BioProject database under accession code PRJNA998474. Phosphorylation database in this study can be found in this paper[25](https://doi.org/10.1038/s41467-021-24816-z). Source data are provided as a Source Data file. Source data are provided with this paper.

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

## Acknowledgements

We thank members of Mello lab for discussions, Caenorhabditis Genetics Center (CGC) for providing strain, Dr. Weifeng Gu for providing the PIR-1 protein for preparing small RNA sequencing libraries. This work was supported by the following grants to C.C.M.: NIH grants (GM058800 and HD078253), C.C.M. is a Howard Hughes Medical Institute Investigator.

## Author contributions

G.M.Z. and C.C.M. designed the experiments. G.M.Z. and Y.H.D performed the experiments. C.W.Z analyzed the sequencing results. G.M.Z. and C.C.M. wrote the manuscript.

## Competing interests

The authors declare no competing interests.
