## [Peer Review File · Nature Communications]

Casein kinase II promotes piRNA production through direct phosphorylation of USTC component TOFU-4REVIEWER COMMENTS

Reviewer #1 (Remarks to the Author):

Zhang et al. reports the casein kinase II (CK2) as a novel factor for the assembly of the upstream sequence transcription complex (USTC) and piRNA transcription. Using a reverse genetic approach, the authors identify that CK2 is required for piRNA-mediated silencing. CK2 promotes PIWI protein PRG-1 expression and USTC piRNA focus formation. Previous studies reported that CK2 mediates protein phosphorylation to affect protein-protein interaction. Using co-IP (in vivo) and phosphorylation assays (in vitro), the authors investigate that CK2 interacts with TOFU-4 and specifically phosphorylates TOFU-4 to promote the assembly of USTC. Meanwhile, Figure 6 proves that CK2 affects PRG-1 and USTC in the aging process.

This paper is a very complete analysis of how CK2 participates in piRNA gene transcription. I recommend publication in current form. Besides I have several minor queries.

In Figure 3B, why free GFP became prominent when there is an intact GFP::PRG-1 translation fusion?

TOFU-4 is a component of the USTC complex and KIN-3 was identified as a yeast two-hybrid (Y2H) binding partner of TOFU-4. We wonder whether there are other unidentified USTC factors or UAD-2 or other integrator factors also be targeted by CK2?

The RNAi screen was performed against all 947 genes in the embryo lethal subset of the *C. elegans* RNAi collection (Ahringer). A list of the 947 genes or a citation of the list should be included.

Reviewer #2 (Remarks to the Author):

The manuscript by Zhang et al addresses the role in CK2 (of which KIN-3 is a key subunit in *C. elegans*) in the worm piRNA pathway. They find that TOFU-4, a subunit of a transcription complex that acts at piRNA loci, is a substrate of CK2 and that phosphorylation of TOFU-4 helps the formation of the typical transcription foci at piRNA loci, and is important for silencing a piRNA sensor. They also show that during ageing TOFU-4 phosphorylation drops, in parallel to a decline of CK2 activity, and accompanied by loss of piRNA silencing efficiency.

Overall, this is a very nice manuscript, that bring to light the PTM-aspects of the piRNA pathway in *C. elegans*. This is uncharted territory, and the reported findings make a strong case for the proposed model. I support publication in Nature Comm., but do think that the authors should improve a number of aspects of their work before the work can really be accepted for publication.

Main points:

1. The authors successfully identified potential TOFU-4 phosphorylation sites and supported their findings with in vitro data. However, to strengthen their conclusions, it is essential for the authors to demonstrate the difference in phosphorylation levels between wild-type TOFU-4 and the TOFU-4 (S131A S229A) mutant in vivo, for example, by phos-tag gels. This will help account for other factors that may influence phosphorylation (is TOFU-4 possibly phosphorylated at other sites?)
2. The authors observed that KIN-3 coimmunoprecipitated TOFU-4 and according to the previous Y2H data, this interaction is direct. To test whether this interaction may be responsible for TOFU-4 phosphorylation, it is important to test if the TOFU-4 (S131A S229A) mutant can still interact with KIN-3 or not. Does the observed interaction reflect substrate binding (in which case it should be lost) or another mode of interaction?
3. The study shows that TOFU-4 (S131A S229A) has reduced co-precipitation with PRDE-1 compared to wild-type TOFU-4. To strengthen the idea that TOFU-4 phosphorylation status drives this interaction, the authors should investigate the effect of the phosphomimetic mutant TOFU-4 (S131D S229D) on co-precipitation with PRDE-1. Also Y2H testing of such mutants would be interesting to report.
4. The authors refer to the *C. elegans* phosphoproteome dataset that used a mixed worm population for the phosphoproteome. The current study should ideally show the changes in phosphorylation level of USTC proteins after kin-10 and kin-3 depletion using (for instance) phos-tag gels (in young adults).
5. It would be beneficial to include mass-spectrum data for TOFU-4 (S131A S229A) co-immunoprecipitation. This could provide a broader view of the interactions involved and offer more insights into the specificity of the observed interaction (changes).
6. When reporting on sequencing, the authors write that 2 samples were sequenced. Yet, their plots show error bars. However, no statistics are possible with n=2. Plotting both replicates next to each other would be the proper way to show the data. Better would be to take n>2 and do proper statistics. However, with the strong effects reported, this could be considered unnecessary. In case the error bars reflect the analysis of the many different piRNA loci this is also not correct, as the error bar would not reflect on experimental variation but on variability of expression of individual piRNA loci (which is not the aim of the study).
7. Precursor and mature piRNA sequencing should be done for the phosphorylation mutants, and the ageing. Sensor activation could also have a different cause, and sequencing would address that concern.
8. In general, the imaging of any situation is restricted to a single image. No sense of variability can be deduced from this. Some form of quantification has to be presented.

9. How does KIN-3:GFP look in *tofu-4* mutants? Does it go away from eth presumed chromatin? Is there any colocalization between KIN-3 and PRDE-1?

10. In the phospho mimic mutant that was made both serines were mutated. However, there is no data to show that this is the relevant situation. Single mutations should be tested before anything can be concluded. The double mimic could in some unforeseen way erase the effect of single events. Is the phospho mimic in PRDE-1 foci?

Minor:

Please annotate the pachytene zone in 1A,B. For non-elegans researchers the presented information is too scarce.

Please provide reference for *line2h* and *turmoil1* activation in piRNA mutants (page 5, bottom)

Please show the recombinant proteins purified from *E. coli*. The readers need to know about the quality of the protein preps used.

In the IP-Westerns, please provide information about quantities (how much was used for input in relation to the IP samples?)

Testing CK2 activity in extracts seems a bit of a crude assay, given that many more kinases exist. Can a control be provided using extract from a *kin-3* mutant (using the AID system that the authors developed)?

Page 13: '...promotes the subnuclear localization...' It is unclear to me what is meant. Not much is known about the subnuclear localization of PRDE-1 foci, apart from that there are foci. So what can be 'promoted'? I guess the author aim at potential colocalization of these foci with nuclear pores? Please explain better what is meant.

We would like to thank the reviewers for their insightful and constructive feedback. Using these comments as a guide, we have revised our manuscript. Our revisions encompass the integration of new data and we have made extensive revisions to both the results and the discussion sections. These changes include performing the following additional experiments:

- We performed phos-tag gel experiments to detect the phosphorylation (**supplementary Figure 4a**).
- We conducted the pull-down assay to examine whether TOFU-4 (S131A S229A) mutant retains its ability to interact with KIN-3 (**supplementary Figure 4d**).
- We conducted the co-immunoprecipitation (co-IP) assay to compare the interaction capacity between the wild type TOFU-4 and the phospho mimetic mutant TOFU-4 (S131D S229D), and also we checked the localization of PRDE-1 foci in wild type and *tofu-4* (S131D S229D) mutants (**Figure 5b-e**).
- We performed the small RNA sequencing to detect the precursor and mature piRNA in phosphorylation mutants and in aging (**Figure 4f, g and Figure 6g, h**).
- We examined KIN-3:GFP localization in *tofu-4* mutants and the colocalization between KIN-3 and PRDE-1 (**supplementary Figure 2d, e**).

Here are our point-by-point responses to the reviewers' comments.

Reviewer #1 (Remarks to the Author):

Zhang et al. reports the casein kinase II (CK2) as a novel factor for the assembly of the upstream sequence transcription complex (USTC) and piRNA transcription. Using a reverse genetic approach, the authors identify that CK2 is required for piRNA-mediated silencing. CK2 promotes PIWI protein PRG-1 expression and USTC piRNA focus formation. Previous studies reported that CK2 mediates protein phosphorylation to affect protein-protein interaction. Using co-IP (in vivo) and phosphorylation assays (in vitro), the authors investigate that CK2 interacts with TOFU-4 and specifically phosphorylates TOFU-4 to promote the assembly of USTC. Meanwhile, Figure 6 proves that CK2 affects PRG-1 and USTC in the aging process.

This paper is a very complete analysis of how CK2 participates in piRNA gene transcription. I recommend publication in current form. Besides I have several minor queries.

We thank the reviewer for the positive comments on our work.

1) In Figure 3B, why free GFP became prominent when there is an intact GFP::PRG-1 translation fusion?

We apologize for not explaining this result better. We have revised the paragraph on this finding as follows to address the reviewer's question:

“Previous studies have shown that both PRG-1 protein levels and its localization to peri-nuclear nuage, P granules, are reduced in mutants that disrupt expression of

piRNA precursors, while PRG-1 mRNA levels remain unaffected¹. These findings suggest that PRG-1 like some other Argonaute proteins such as Ago2 becomes unstable when unloaded². Consistent with these previous findings we found that in *kin-3* depleted animals, PRG-1 mRNA levels were not reduced compared to wild type (Supplementary Fig. 2a), while in contrast, GFP::PRG-1 failed to localize in P-granules and instead localized diffusely in the cytosol (Fig. 3a). We observed an identical change in GFP::PRG-1 localization in animals with mutations in *tofu-4* (Fig. 3a), a gene previously shown to be required for piRNA transcription and for PRG-1 protein stability³. When assayed by western blotting with GFP-specific antibodies in *kin-3* and *tofu-4* mutants, the band corresponding to full-length GFP::PRG-1 protein was strongly reduced and a 30KD band, corresponding in size to free GFP, became prominent (Fig. 3b). This finding suggests that the N-terminal GFP tag is proteolytically released from GFP::PRG-1 during the PRG-1 instability induced by *kin-3* and *tofu-4* mutations, and suggest that persistence of this proteolytic fragment explains the cytoplasmic GFP fluorescence observed in the mutants.”

2) *TOFU-4 is a component of the USTC complex and KIN-3 was identified as a yeast two-hybrid (Y2H) binding partner of TOFU-4. We wonder whether there are other unidentified USTC factors or UAD-2 or other integrator factors also be targeted by CK2?*

We have modified the discussion to address this possibility:

Discussion Pages 13: “A search within the phosphorylation database⁴ focusing on the USTC factors PRDE-1, SNPC-4, TOFU-4 and TOFU-5, in addition to the recently identified chromodomain factor UAD-2 which localizes to piRNA cluster chromatin⁵, revealed that among these factors, TOFU-4 is the sole protein displaying detectable phosphorylation. While there could be additional piRNA biogenesis factors like UAD-2 subject to CK2 phosphorylation, our study suggests that posttranslational modification of TOFU-4 by CK2 is important for assembly of the piRNA biogenesis machinery and for piRNA pathway function.”

3) *The RNAi screen was performed against all 947 genes in the embryo lethal subset of the C. elegans RNAi collection (Ahringer). A list of the 947 genes or a citation of the list should be included.*

As suggested by the reviewer, we have incorporated the list of the 945 tested genes into the updated manuscript (**Table 1**).

Reviewer #2 (Remarks to the Author):

The manuscript by Zhang et al addresses the role in CK2 (of which KIN-3 is a key subunit in C. elegans) in the worm piRNA pathway. They find that TOFU-4, a subunit

of a transcription complex that acts at piRNA loci, is a substrate of CK2 and that phosphorylation of TOFU-4 helps the formation of the typical transcription foci at piRNA loci, and is important for silencing a piRNA sensor. They also show that during ageing TOFU-4 phosphorylation drops, in parallel to a decline of CK2 activity, and accompanied by loss of piRNA silencing efficiency.

Overall, this is a very nice manuscript, that bring to light the PTM-aspects of the piRNA pathway in *C. elegans*. This is uncharted territory, and the reported findings make a strong case for the proposed model. I support publication in *Nature Comm.*, but do think that the authors should improve a number of aspects of their work before the work can really be accepted for publication.

We thank the reviewer for the thoughtful and constructive comments.

Main points:

1. The authors successfully identified potential TOFU-4 phosphorylation sites and supported their findings with *in vitro* data. However, to strengthen their conclusions, it is essential for the authors to demonstrate the difference in phosphorylation levels between wild-type TOFU-4 and the TOFU-4 (S131A S229A) mutant *in vivo*, for example, by phos-tag gels. This will help account for other factors that may influence phosphorylation (is TOFU-4 possibly phosphorylated at other sites?)

We thank the reviewer for this suggestion, we performed phos-tag analysis on TOFU-4 but no mobility shift was detected, even on protein phosphorylated by CK2 *in vitro*. Phosphorylation does not always lead to a change in protein mobility⁶. We have added the following to the results section to address these new studies: “In some instances phosphorylation of a protein can cause a shift in mobility on an SDS gel⁶. To ask if phosphorylation of TOFU-4 by CK2 *in vitro*, or *in vivo* alters TOFU-4 mobility we used the previously published phos-tag gel method⁷. However, no significant shift was observed even for the *in vitro* phosphorylated TOFU-4 (Supplementary Fig. 4a).”

2. The authors observed that KIN-3 coimmunoprecipitated TOFU-4 and according to the previous Y2H data, this interaction is direct. To test whether this interaction may be responsible for TOFU-4 phosphorylation, it is important to test if the TOFU-4 (S131A S229A) mutant can still interact with KIN-3 or not. Does the observed interaction

reflect substrate binding (in which case it should be lost) or another mode of interaction?

As suggested by the reviewer, we conducted the pull-down assay to examine whether TOFU-4 (S131A S229A) mutant retains its ability to interact with KIN-3. We found that TOFU-4 (S131A S229A) still interacts with KIN-3. Consequently, we think that CK2 phosphorylation of TOFU-4 does not influence the binding between TOFU-4 and KIN-3, but rather affects the assembly of PRDE-1 foci. The results have been included in the **supplementary Figure 4d** and are discussed on **Page 13** as follows: “Mechanistically, our findings suggest that CK2 functions upstream of USTC assembly. TOFU-4 is a direct CK2 substrate and mutation of presumptive CK2 phosphorylation sites in TOFU-4 did not affect its interaction with KIN-3 (Supplementary Fig. 4d) but instead reduced the association of TOFU-4 with the USTC component PRDE-1.”

3. The study shows that TOFU-4 (S131A S229A) has reduced co-precipitation with PRDE-1 compared to wild-type TOFU-4. To strengthen the idea that TOFU-4 phosphorylation status drives this interaction, the authors should investigate the effect of the phosphomimetic mutant TOFU-4 (S131D S229D) on co-precipitation with PRDE-1. Also Y2H testing of such mutants would be interesting to report.

We thank the review for these suggestions. We conducted the *in vivo* studies. Our results suggested that, while the alanine substitutions abolished the interaction, the phospho mimetic mutant TOFU-4 (S131D S229D) still exhibited co-precipitation with PRDE-1. We noted, however, a 40% reduction in the co-IP, suggesting that the aspartic acid residues do not fully mimic phosphorylation, a finding consistent with the intermediate effect on silencing (**Supplementary Fig. 4b**). Moreover, consistent with these findings we now include localization studies showing that compared to the wild type protein, TOFU-4 (S131D S229D) protein is still co-localized with PRDE-1 foci, albeit with smaller foci. These results have been included in the main **Figure 5b-e**.

4. The authors refer to the *C. elegans* phosphoproteome dataset that used a mixed worm population for the phosphoproteome. The current study should ideally show the changes in phosphorylation level of USTC proteins after *kin-10* and *kin-3* depletion using (for instance) *phos-tag* gels (in young adults).

This has been done. See our response to question 1.

5. It would be beneficial to include mass-spectrum data for TOFU-4 (S131A S229A) co-immunoprecipitation. This could provide a broader view of the interactions involved and offer more insights into the specificity of the observed interaction (changes).

We thank the reviewer for this valuable suggestion which could no doubt lead to many future lines of study. While time and resources do not permit a full-scale proteomic study on TOFU-4 we have taken a candidate approach to broaden our understanding of how TOFU-4 (S131A S229A) impacts the co-assembly of other USTC factors. In line with our results, we found that SNPC-4 foci are disrupted in *tofu-4(S131A S229A)* mutants as well.

6. When reporting on sequencing, the authors write that 2 samples were sequenced. Yet, their plots show error bars. However, no statistics are possible with $n=2$. Plotting both replicates next to each other would be the proper way to show the data. Better would

be to take $n > 2$ and do proper statistics. However, with the strong effects reported, this could be considered unnecessary. In case the error bars reflect the analysis of the many different piRNA loci this is also not correct, as the error bar would not reflect on experimental variation but on variability of expression of individual piRNA loci (which is not the aim of the study).

We apologize for our careless use of the error bars. We have corrected this. These results have been included in the main **Figure 2b, c and supplementary Figure 1**.

7. Precursor and mature piRNA sequencing should be done for the phosphorylation mutants, and the ageing. Sensor activation could also have a different cause, and sequencing would address that concern.

As suggested by the reviewer, we performed the small RNA sequencing. Consistent with our sensor reporter, we found that in phosphorylation mutants and in aging, both the precursor and mature piRNA are decreased. These results have been included in the main **Figure 4f, g and Figure 6g, h**.

Results Page 10: “To ask if CK2 mediated phosphorylation of TOFU-4 affects piRNA production, we performed small RNA sequencing. We observed that the levels of both mature piRNAs and of piRNA precursors were reduced in the *tofu-4(S131A S229A)* mutants (Fig. 4f, g).”

Results Page 12: “Consistently, we found the mature piRNA and the piRNA precursor

levels were reduced in the aging process (Fig. 6g, h).”

8. *In general, the imaging of any situation is restricted to a single image. No sense of variability can be deduced from this. Some form of quantification has to be presented.*

We have clearly stated the number and percentage in the paper. Specifically, for **Figures 3a, 3c, 3d, 3e, 5c, 5d and 5e**, more than 50 worms are counted, the phenotype is similar. For **Figure 6**, the number and percentage are labeled in the figure.

9. *How does KIN-3:GFP look in tofu-4 mutants? Does it go away from eth presumed chromatin? Is there any colocalization between KIN-3 and PRDE-1?*

As suggested by the reviewer, we examined KIN-3:GFP localization in *tofu-4* mutants, but did not observe significant change.

Since KIN-3:GFP is widely expressed in the cell, we can see the colocalization between KIN-3 and PRDE-1. However, we did not detect any enriched signal of KIN-3::GFP in PRDE-1 foci. The results have been included in the **supplementary Figure 2d, e**.

Results Page 8: “Indeed, we can see the colocalization between KIN-3 and PRDE-1. However, we did not detect any enriched signal of GFP::KIN-3 in PRDE-1 foci. Additionally, we did not identify any changes of GFP::KIN-3 localization in *tofu-4* mutants (Supplementary Fig. 2d, e).”

10. *In the phospho mimic mutant that was made both serines were mutated. However, there is no data to show that this is the relevant situation. Single mutations should be tested before anything can be concluded. The double mimic could in some unforeseen way erase the effect of single events. Is the phospho mimic in PRDE-1 foci?*

As suggested by the reviewer, we tested the single mutations. The S131D and S229D lesions each caused partial de-silencing individually.

We also examined the *tofu-4* phospho mimetic mutant. We found that like the wild type protein, the mutant protein is still localized in the PRDE-1 foci. However, the PRDE-1 foci are a little smaller. The results are included in the main **Figure 5e** and in the **supplementary Figure 4b**.

Minor:

Please annotate the pachytene zone in 1A,B. For non-elegans researchers the presented information is too scarce.

As suggested by the reviewer, we have annotated the pachytene zone in 1A, B. The results have been included in the main **Figure 1a and b**.

Please provide reference for *line2h* and *turmoil1* activation in piRNA mutants (page 5, bottom)

We have provided reference⁸ for *line2h* and *turmoil1* activation in piRNA mutants in the paper.

Please show the recombinant proteins purified from *E. coli*. The readers need to know about the quality of the protein preps used.

As suggested by the reviewer, the recombinant proteins purified from *E. coli* are shown in **supplementary Figure 3a**.

In the IP-Westerns, please provide information about quantities (how much was used for input in relation to the IP samples?)

We have provided this information in the main **Figure 4a and 5b**.

Testing CK2 activity in extracts seems a bit of a crude assay, given that many more kinases exist. Can a control be provided using extract from a *kin-3* mutant (using the AID system that the authors developed)?

We provided the control using the extract from *kin-3* mutant, this has been included in **supplementary Figure 4c**.

Page 13: ‘...promotes the subnuclear localization...’ It is unclear to me what is meant. Not much is known about the subnuclear localization of PRDE-1 foci, apart from that

there are foci. So what can be 'promoted'? I guess the author aim at potential colocalization of these foci with nuclear pores? Please explain better what is meant.

As suggested by the reviewer, we have changed the sentences “In the *C. elegans* pachytene germline PRDE-1 foci colocalize with large regions of chromosome IV where the majority of piRNA genes reside and become positioned near the nuclear periphery⁹. The transcription of piRNA genes is thought to occur within chromatin marked by H3K27me3, a mark also found in telomeric chromatin¹⁰. It will be interesting to learn whether CK2 phosphorylation of TOFU-4 promotes the colocalization of the huge megabase-scale clusters of piRNA genes perhaps to coordinate piRNA transcription with piRNA export and processing.”

References:

1. Belicard, T., Jareosettasin, P. & Sarkies, P. The piRNA pathway responds to environmental signals to establish intergenerational adaptation to stress. *BMC Biol* **16**, 103 (2018).
2. Smibert, P., Yang, J.-S., Azzam, G., Liu, J.-L. & Lai, E.C. Homeostatic control of Argonaute stability by microRNA availability. *Nature Structural & Molecular Biology* **20**, 789-795 (2013).
3. Weng, C. *et al.* The USTC co-opts an ancient machinery to drive piRNA transcription in *C. elegans*. *Genes Dev* **33**, 90-102 (2019).
4. Li, W.J. *et al.* Insulin signaling regulates longevity through protein phosphorylation in *Caenorhabditis elegans*. *Nat Commun* **12**, 4568 (2021).
5. Huang, X. *et al.* A chromodomain protein mediates heterochromatin-directed piRNA expression. *Proc Natl Acad Sci U S A* **118** (2021).
6. Nishioka, K., Kato, Y., Ozawa, S.I., Takahashi, Y. & Sakamoto, W. Phos-tag-based approach to study protein phosphorylation in the thylakoid membrane. *Photosynth Res* **147**, 107-124 (2021).
7. O'Donoghue, L. & Smolenski, A. Analysis of protein phosphorylation using Phos-tag gels. *J Proteomics* **259**, 104558 (2022).
8. Bagijn, M.P. *et al.* Function, targets, and evolution of *Caenorhabditis elegans* piRNAs. *Science* **337**, 574-578 (2012).
9. Weick, E.-M. *et al.* PRDE-1 is a nuclear factor essential for the biogenesis of Ruby motif-dependent piRNAs in *C. elegans*. *Genes & Development* **28**, 783-796 (2014).
10. Jamieson, K. *et al.* Telomere repeats induce domains of H3K27 methylation in *Neurospora*. *Elife* **7** (2018).

REVIEWER COMMENTS

Reviewer #1 (Remarks to the Author):

The authors have perfectly addressed my comments and questions. I strongly recommend the publication of this work.

Reviewer #2 (Remarks to the Author):

The authors have addressed most issues in their revised manuscript very well. There are a few things that should still be improved in my opinion, but these are textual.

1) In Figure 1 the authors describe the RppH treatment as the method being used for precursor sequencing. Please also clarify this in the methods. In the methods, it simply states PIR-1 or RppH was used, but no further context. Also in the Figure legends to 4g and 6h this should be clarified to help the reader.

2) On the precursor quantification: I now see that these are done only once for each sample. As I do not think it fair to demand new experiments upon a re-review, I would not ask for adding at least one more, but the authors will need to spell out prominently (so not hidden in the methods) that the precursor results are based on an n=1 experiment. Especially in the ageing sample, one may have serious doubts about the significance of the effect, and hence it cannot be presented as is. The text needs to reflect the uncertainty that is inevitably connected to an n=1 experiment. Ideally, the authors shape up the sequencing to what is commonly done: at least 2, better 3 biological replicates.

3) A second issue related to the precursor quantification: it would be much better to simply show a bar plot, like for the mature piRNAs. The precursors in elegans are well defined, and can simply be counted. This is the information that one needs, and in the current representations this is only indirectly detectable-hence not suitable. Showing the nucleotide coverage down to -9 has no information in the current context. It only has relevance if one wants to characterize them, but that has been done already, and is not the focus of the presented experiment. Even if the authors fear that some atypical precursors, that extend beyond -2, may be relevant, a dedicated count for such reads could be presented in a separate bar plot.

4) The last sentence of the results is too strong:

Consequently, reduced CK2 activity serves as the underlying cause of piRNA-mediated gene silencing defects during the aging process.

The authors only show correlations, and do not show causality. For instance, they do not show that the effects can be reversed by increasing CK2 activity in aged animals. Therefore, it cannot be concluded that reduced CK2 activity is the underlying cause. Something like "our results are consistent with reduced CK2 activity being a driving force in aging" would more accurately summarize the findings.

We would like to thank the reviewers for the helpful feedback. Now we have revised our manuscript. Here are our point-by-point responses to the reviewer's comments.

Reviewer #1 (Remarks to the Author):

The authors have perfectly addressed my comments and questions. I strongly recommend the publication of this work.

We thank the reviewer for the positive feedback.

Reviewer #2 (Remarks to the Author):

The authors have addressed most issues in their revised manuscript very well. There are a few things that should still be improved in my opinion, but these are textual.

1) In Figure 1 the authors describe the RppH treatment as the method being used for precursor sequencing. Please also clarify this in the methods. In the methods, it simply states PIR-1 or RppH was used, but no further context. Also in the Figure legends to 4g and 6h this should be clarified to help the reader.

We thank the reviewer for this helpful suggestion. We have clarified that in the methods and in the figure legends to 2e, 4g and 6h.

Methods Page 17: "For mature piRNA sequencing, samples were treated with homemade PIR-1 for 2hs. For piRNA precursor sequencing, samples were pretreated with 5' Pyrophosphohydrolase (RppH, NEB) for 1h, and purified with RNA purification kit, then treated with homemade PIR-1 for 2hs."

Figure legends Pages 23, 25 and 27: "Small RNAs are pretreated with 5' Pyrophosphohydrolase (RppH, NEB)."

2) On the precursor quantification: I now see that these are done only once for each sample. As I do not think it fair to demand new experiments upon a re-review, I would not ask for adding at least one more, but the authors will need to spell out prominently (so not hidden in the methods) that the precursor results are based on an n=1 experiment. Especially in the ageing sample, one may have serious doubts about the significance of the effect, and hence it cannot be presented as is. The text needs to reflect the uncertainty that is inevitably connected to an n=1 experiment. Ideally, the authors shape up the sequencing to what is commonly done: at least 2, better 3 biological replicates.

We apologize for not explaining this result better. We did 2 biological replicates. They are very similar, so we only showed one typical result in the previous manuscript. As suggested by the reviewer, we have shown the 2 biological replicates and changed the precursor quantification to simply show a bar plot in the updated manuscript in the main **Figure 2e, Figure 4g and Figure 6h.**

3) *A second issue related to the precursor quantification: it would be much better to simply show a bar plot, like for the mature piRNAs. The precursors in elegans are well defined, and can simply be counted. This is the information that one needs, and in the current representations this is only indirectly detectable-hence not suitable. Showing the nucleotide coverage down to -9 has no information in the current context. It only has relevance if one wants to characterize them, but that has been done already, and is not the focus of the presented experiment. Even if the authors fear that some a-typical precursors, that extend beyond -2, may be relevant, a dedicated count for such reads could be presented in a separate bar plot.*

This has been done. See our response to question 2.

4) *The last sentence of the results is too strong:
Consequently, reduced CK2 activity serves as the underlying cause of piRNA-mediated gene silencing defects during the aging process.
The authors only show correlations, and do not show causality. For instance, they do not show that the effects can be reversed by increasing CK2 activity in aged animals. Therefore, it cannot be concluded that reduced CK2 activity is the underlying cause. Something like “our results are consistent with reduced CK2 activity being a driving force in aging” would more accurately summarize the findings.*

As suggested by the reviewer, we have corrected the sentences.

Results Page 12: “Taken together, these results are consistent with the possibility that declining CK2 activity contributes to the piRNA-mediated gene silencing defects observed during the aging process.”